# Ferrimagnetic Ordering and Spin-Glass State in Diluted GdFeO_3_-Type Perovskites (Lu_0.5_Mn_0.5_)(Mn_1−*x*_Ti*_x_*)O_3_ with *x* = 0.25, 0.50, and 0.75

**DOI:** 10.3390/ma16041506

**Published:** 2023-02-10

**Authors:** Alexei A. Belik, Ran Liu, Andreas Dönni, Masahiko Tanaka, Kazunari Yamaura

**Affiliations:** 1International Center for Materials Nanoarchitectonics (WPI-MANA), National Institute for Materials Science (NIMS), Namiki 1-1, Tsukuba 305-0044, Ibaraki, Japan; 2Graduate School of Chemical Sciences and Engineering, Hokkaido University, North 10 West 8, Kita-ku, Sapporo 060-0810, Hokkaido, Japan; 3Institute of Scientific and Industrial Research, Osaka University, Mihogaoka 8-1, Ibaraki, Osaka 567-0047, Japan; 4National Institute for Materials Science (NIMS), Sengen 1-2-1, Tsukuba 305-0047, Ibaraki, Japan

**Keywords:** GdFeO_3_-type perovskites, ferrimagnetic, spin glass, crystal structures, structural disorder

## Abstract

ABO_3_ perovskite materials with small cations at the A site, especially those with ordered cation arrangements, have attracted a great deal of interest because they show unusual physical properties and deviations from the general characteristics of perovskites. In this work, perovskite solid solutions (Lu_0.5_Mn_0.5_)(Mn_1−*x*_Ti*_x_*)O_3_ with *x* = 0.25, 0.50, and 0.75 were synthesized by means of a high-pressure, high-temperature method at approximately 6 GPa and approximately 1550 K. All the samples crystallize in the GdFeO_3_-type perovskite structure (space group *Pnma*) and have random distributions of the small Lu^3+^ and Mn^2+^ cations at the A site and Mn^4+/3+/2+^ and Ti^4+^ cations at the B site, as determined by Rietveld analysis of high-quality synchrotron X-ray powder diffraction data. Lattice parameters are *a* = 5.4431 Å, *b* = 7.4358 Å, *c* = 5.1872 Å (for *x* = 0.25); *a* = 5.4872 Å, *b* = 7.4863 Å, *c* = 5.2027 Å (for *x* = 0.50); and *a* = 5.4772 Å, *b* = 7.6027 Å, *c* = 5.2340 Å (for *x* = 0.75). Despite a significant dilution of the A and B sublattices by non-magnetic Ti^4+^ cations, the *x* = 0.25 and 0.50 samples show long-range ferrimagnetic order below *T*_C_ = 89 K and 36 K, respectively. Mn cations at both A and B sublattices are involved in the long-range magnetic order. The *x* = 0.75 sample shows a spin-glass transition at *T*_SG_ = 6 K and a large frustration index of approximately 22. A temperature-independent dielectric constant was observed for *x* = 0.50 (approximately 32 between 5 and 150 K) and for *x* = 0.75 (approximately 50 between 5 and 250 K).

## 1. Introduction

ABO_3_ perovskite materials usually have relatively large cations at the A site [1] with one of the smallest typical cations being Lu^3+^ with ionic radius *r*_XIII_(Lu^3+^) = 0.977 Å [2]. Perovskites containing smaller cations at the A site are sometimes called “exotic” [3,4,5,6], and they usually require a high-pressure, high-temperature method for their synthesis. Such exotic perovskites include InBO_3_ (with *r*_XIII_(In^3+^) = 0.92 Å [2]), ScBO_3_ (with *r*_XIII_(Sc^3+^) = 0.870 Å [2]) and MnBO_3_ (with *r*_XIII_(Mn^2+^) = 0.96 Å [2]). They show unusual physical properties and deviations from general perovskite tendencies [3]. The reasons for this are the magnetism of 3d^5^ cations (Mn) at the A sites and large structural distortions/tilts of the BO_6_ octahedral framework caused by small cations at the A sites (for A = In, Sc and Mn). For example, RCrO_3_ compounds with rare-earth elements R = La-Lu adopt G-type antiferromagnetic (AFM) structures, while InCrO_3_ and ScCrO_3_ already have C-type AFM structures [4,7]. Some MnBO_3_ compounds exhibit complex incommensurate magnetic structures at the A sites and other interesting physical properties [8].

(R^3+^_1−*y*_Mn^2+^*_y_*)BO_3_ solid-solution perovskites can also move into the “exotic” region when the average ionic radius at the A site becomes smaller than that of Lu^3+^. The introduction of 3d^5^ cations (Mn^2+^) into the A sites can significantly modify the physical properties of (R_1−*y*_Mn*_y_*)MnO_3_ solid solutions [9,10,11,12,13] and cause them to differ from their parent RMnO_3_ compounds. For example, a small percentage of Mn^2+^ at the A site of (Tm_1−*y*_Mn*_y_*)MnO_3_ [11] stimulates the long-range magnetic ordering of all A-site Tm^3+^ and Mn^2+^ cations at a higher temperature than Tm^3+^ in TmMnO_3_ [14], promotes ferrimagnetic structures (instead of AFM ones), and results in a negative magnetization phenomenon based on the Néel model for N-type ferrimagnets [15]. The range of the chemical composition of (R_1−*y*_Mn*_y_*)MnO_3_ solid solutions shrinks with the increase in the size of R^3+^ cations [9,10,11,12,13], and the (R_1−*y*_Mn*_y_*)MnO_3_ solid solutions are formed for 0 ≤ *y* ≤ 0.4 [10] for the smallest R^3+^ cation R = Lu.

We found that the compositional range of (R_1−*y*_Mn*_y_*)MnO_3_ solid solutions can be extended to beyond *y* = 0.4 through an additional doping at the B sites in “double” solid solutions (R_1−*y*_Mn*_y_*)(Mn_1−*x*_Ti*_x_*)O_3_ [16]. For example, the doping level can reach *y* ≈ 0.8 for *y* = *x*. There are several variable parameters for (R_1−*y*_Mn*_y_*)(Mn_1−*x*_Ti*_x_*)O_3_, such as *x*, *y*, R, and synthesis conditions. In this work, we report the synthesis, the crystal structures, and the physical properties of such “double” solid solutions with R = Lu and *y* = 0.5. We selected the half-doped level at the A sites (*y* = 0.5) in order to compare such perovskites, where R and Mn atoms are disordered in one site, with A-site columnar-ordered quadruple perovskites R_2_MnMn(Mn_4−*z*_Ti*_z_*)O_12_ (the generic composition is A_2_A′A″B_4_O_12_ [17]), where R and Mn atoms are ordered [18,19,20], to observe the effects of cation ordering on magnetic properties. In addition, we selected a non-magnetic R^3+^ cation to eliminate effects of rare-earth magnetism [16]. We observed that magnetic properties were qualitatively similar for *x* = 0.25 and 0.50 (*z* = 1 and 2). On the other hand, magnetic properties were quite different for *x* = 0.75 (*z* = 3), where structural disorder at the A sites produces spin-glass magnetic states, while ordering of R and Mn atoms produces ferrimagnetic structures.

## 2. Experimental

The commercial chemicals Lu_2_O_3_ (99.9%) and TiO_2_ (99.9%) and homemade Mn_2_O_3_ (99.99%) and MnO (99.99%) in stoichiometric amounts were used in initial oxide mixtures for the preparation of solid solutions of (Lu_0.5_Mn_0.5_)(Mn_1−*x*_Ti*_x_*)O_3_ with *x* = 0.25, 0.50, and 0.75. The synthesis was performed using a high-pressure, high-temperature solid-state method in Au capsules. A NIMS belt-type high-pressure apparatus was used. The synthesis was performed at approximately 6 GPa and 1550 K for 2 h. After annealing at the synthesis temperature, the heating current was turned off, which resulted in rapid cooling (quenching) of the samples; thereafter, the pressure was reduced to ambient pressure over approximately 40 min.

X-ray powder diffraction (XRPD) data were measured at room temperature (RT) with a MiniFlex600 diffractometer (Rigaku, Tokyo, Japan). The measurement conditions were CuKα radiation, a 2*θ* range of 8–100°, a step of 0.02°, and scan speed of 1 °/min. Synchrotron XRPD data were collected at RT on the BL15XU beamline (the former NIMS beamline) of SPring-8 [21]. The measurement conditions were *λ* = 0.65298 Å, a 2*θ* range of 2.04–60.23°, a step of 0.003°, and a measurement time of 30–60 s; Lindemann glass capillary tubes with an inner diameter of 0.1 mm were used and were rotated during measurements. The Rietveld analysis was performed using the *RIETAN-2000* program [22].

Energy-dispersive X-ray (EDX) spectra and scanning electron microscopy (SEM) images were obtained on a Miniscope TM3000 (Hitachi, Tokyo, Japan) working at 15 kV.

Temperature-dependent magnetic measurements were performed between 2 and 400 K in applied fields of 100 Oe and 10 kOe using MPMS-XL-7T and MPMS3 SQUID magnetometers (Quantum Design, San Diego, CA, USA) under both zero-field-cooled (ZFC) and field-cooled on cooling (FCC) conditions. The isothermal magnetic field dependence was measured at different temperatures between −70 and 70 kOe. Frequency-dependent alternating current (ac) susceptibility measurements were performed on cooling with MPMS-1T and MPMS3 instruments (Quantum Design, San Diego, CA, USA) at different frequencies (*f*), with different applied oscillating magnetic fields (*H*_ac_), and at zero static dc field (*H*_dc_ = 0 Oe).

A commercial PPMS calorimeter (Quantum Design, San Diego, CA, USA) utilizing a pulse relaxation method was used for specific heat (*C*_p_) measurements, which were recorded from 300 K to 2 K at zero magnetic field and from 150 K (or 200 K) to 2 K at a magnetic field of 90 kOe. All of the magnetic and specific heat measurements were performed using pieces of pellets.

An Alpha-A High Performance Frequency Analyzer (NOVOCONTROL Technologies, Montabaur, Germany) was used to record the dielectric property measurements of the *x* = 0.50 and 0.75 samples. The measurements were performed in a frequency range from 100 Hz to 665 kHz, in a temperature range from 5 K to 330 K (on cooling and heating), and at zero magnetic field. A pellet of the *x* = 0.25 sample was too fragile to withstand polishing and the deposition of electrodes. Therefore, no dielectric measurements were performed for the *x* = 0.25 sample.

## 3. Results and Discussion

No impurities were detected in the *x* = 0.25 and 0.50 samples; this suggested that these were single-phase samples within the sensitivity of the diffraction methods that were used. On the other hand, the *x* = 0.75 sample contained approximately 4.3 weight % of Lu_2_Ti_2_O_7_ impurity (the weight fraction of the impurity was estimated from refined scale factors during the Rietveld analysis). In general, the appearance of impurities suggests that the compositions of main phases should shift slightly from ideal target values. Figure 1 shows the morphology (SEM images) of the obtained samples. The chemical compositions determined by EDX were close to the expected values; the Lu:Mn:Ti ratios were 1.9(2):5.1(1):1.0(1) for *x* = 0.25, 1.9(2):4.0(1):2.1(1) for *x* = 0.50, and 1.8(2):3.0(2):3.2(2) for *x* = 0.75. The large errors could be caused by weak X-ray intensities, which in turn could originate from the charge-up problem.

Structure parameters of (Lu_1−*y*_Mn*_y_*)MnO_3_ solid solutions [10] with space group *Pnma* were used as the initial models for the refinements of the crystal structures of the (Lu_0.5_Mn_0.5_)(Mn_1−*x*_Ti*_x_*)O_3_ samples with *x* = 0.25, 0.50, and 0.75. As in other (R_1−*y*_Mn*_y_*)MnO_3_ systems [9,10,11,12,13], a noticeable anisotropic broadening of reflections was observed in (Lu_0.5_Mn_0.5_)(Mn_1−*x*_Ti*_x_*)O_3_ (inset of Figure 2). Therefore, the introduction of anisotropic broadening corrections significantly improved the fitting results.

It is inaccurate to refine the occupation factors of Ti and Mn at the B sites, as Ti^4+^ and Mn*^n^*^+^ do not differ greatly in their number of electrons. Therefore, the occupation factors at the B sites were fixed at the nominal compositions. On the other hand, the occupation factors of Lu and Mn at the A sites could be refined because Lu^3+^ and Mn^2+^ differed significantly in their number of electrons. Using the constraint *g*(Lu) + *g*(Mn) = 1, the refined occupation factor *g*(Lu) was 0.494(2), 0.506(2), and 0.5318(14) for the *x* = 0.25, 0.50, and 0.75 samples, respectively. These values were close to the nominal values. Therefore, we used the nominal value (*g*(Lu) = 0.5) in the final models.

The refined structural parameters and primary bond lengths and angles in (Lu_0.5_Mn_0.5_)(Mn_1−*x*_Ti*_x_*)O_3_ are summarized in Table 1 and Table 2. Experimental, calculated, and difference synchrotron XRPD patterns are shown in Figure 2 for the *x* = 0.50 sample as an example, where the inset illustrates the absence of any detectable reflections of impurities. Figure 3a shows the crystal structure plotted using the *VESTA* program [23].

The bond valence sum (BVS) values of Lu^3+^ and Mn^2+^ cations [24] at the A sites were close to the expected values of +3 and +2, respectively (Table 2). The BVS values at the B sites, calculated using the average *R*_0_ parameters [24], were very close to the expected value of +3.5.

The magnetic transition temperatures of the three compounds were determined from the peak positions of the differential d(*χT*)/d*T* versus *T* curves measured at a small magnetic field of 100 Oe (Figure 4) as *T*_C_ = 89 K (*x* = 0.25), *T*_C_ = 36 K (*x* = 0.50), and *T*_SG_ = 6 K (*x* = 0.75). *T*_C_ is a ferrimagnetic Curie temperature and *T*_SG_ a spin-glass (SG) temperature. Figure 4 also shows that the magnetic transition temperatures were shifted to higher temperatures (in *x* = 0.25 and 0.50) with the increase in a magnetic field. The magnetic susceptibility curves *χ* versus *T* of the *x* = 0.25 and 0.50 samples were qualitatively similar to each other (Figure 5). The ZFC *χ* versus *T* curves measured at a small magnetic field of 100 Oe showed broad peaks when approaching *T*_C_. The FCC *χ* versus *T* curves (at 100 Oe) showed a sharp increase in susceptibilities below *T*_C_ down to 50 K (*x* = 0.25) and 26 K (*x* = 0.50), and then a decrease down to 2 K. The ZFC and FCC *χ* versus *T* curves (at 100 Oe) showed a clear and strong divergence below approximately *T*_C_. Both ZFC and FCC *χ* versus *T* curves were strongly suppressed when measured at 10 kOe in comparison with the 100 Oe field. Isothermal magnetization, *M* versus *H*, curves of the *x* = 0.25 and 0.50 samples (Figure 6) showed well-defined hysteresis below *T*_C_. Clear jumps of magnetization (on *M* versus *H* curves, at for example *T* = 5 K) near zero field in the *x* = 0.25 sample originate from domain structure changes. *M* versus *H* curves of the *x* = 0.25 sample at different temperatures (Figure 6c) clearly showed that the maximum magnetization is realized near 60 K even at high magnetic fields. The inverse magnetic susceptibilities (*χ*^−1^ versus *T*) deviated from the linear Curie-Weiss law far above *T*_C_ (Figure 7). All these features are typical for materials with long-range ferrimagnetic ordering. The *χ*^−1^ versus *T* curves (FCC, 10 kOe) of the *x* = 0.25 and 0.50 samples could be well fitted by a ferrimagnetic model [25,26] with the equation and obtained fitting parameters given in Figure 7. The opposite signs of the *θ*_1_ and *θ*_2_ parameters could support ferrimagnetic ordering.

On the other hand, the *χ* versus *T* curves of the *x* = 0.75 sample were principally different from those of the *x* = 0.25 and 0.50 samples (Figure 8). The effect of magnetic fields on the susceptibility values of the *x* = 0.75 sample was much weaker. The *χ* versus *T* curve (FCC, 100 Oe) showed a small kink below *T*_SG_ and not a sharp increase. The *M* versus *H* curve at *T* = 5 K showed no detectable hysteresis (because 5 K is very close to *T*_SG_), while a tiny extended S-shaped hysteresis opened at the lower temperature *T* = 2 K (Figure 6b). All these features and the type of divergence between the ZFC and FCC *χ* versus *T* curves at 100 Oe are typical for spin glasses [27,28].

The *χ*^−1^ versus *T* curves (FCC, 10 kOe) could be well fitted by the Curie-Weiss law at high temperatures of 250–400 K (Figure 5 and Figure 8; the fits and fitting parameters are reported in the figures). For the three samples, the experimental effective magnetic moments were close to the calculated ones. This fact supports the ideal Mn charges as (Lu^3+^_0.5_Mn^2+^_0.5_)(Mn^3+^_0.50_Mn^4+^_0.25_Ti^4+^_0.25_)O_3_, (Lu^3+^_0.5_Mn^2+^_0.5_)(Mn^3+^_0.50_Ti^4+^_0.50_)O_3,_ and (Lu^3+^_0.5_Mn^2+^_0.5_)(Mn^2+^_0.25_Ti^4+^_0.75_)O_3_. From the obtained Weiss temperatures, the so-called frustration ratio, defined as |θ|/*T*_C_ or |θ|/*T*_SG_, can be calculated as 1.1 (*x* = 0.25), 4 (*x* = 0.50), and 22 (*x* = 0.75). A very large frustration ratio in the *x* = 0.75 sample indicates a strong degree of magnetic frustration.

To arrive at a deeper understanding of magnetic behavior, we measured ac magnetic susceptibility curves (Figure 9 and Figure 10). The *x* = 0.25 and 0.50 samples showed sharp peaks on both the χ′ versus *T* and the χ″ versus *T* curves. Moreover, there was strong dependence of the χ′ and χ″ values on the applied *H*_ac_ field (insets of Figure 9). These results indicate strong interactions of the *H*_ac_ field with domain structures in these compounds. The appearance of domain structures confirmed the presence of long-range magnetic ordering. On the other hand, no *H*_ac_ field dependence was observed in the *x* = 0.75 sample (inset of Figure 10). There were characteristic shifts of peak positions and intensities as a function of frequency for both the χ′ versus *T* and the χ″ versus *T* curves. With increasing frequency, the peak intensity was reduced for the χ′ versus *T* curves and enhanced for the χ″ versus *T* curves. The SG temperature shift per frequency decade, defined as Δ*T*_SG_/[*T*_SG_Δlog(*f*)], was calculated to be 0.012. For the calculation, we used *T*_SG_ = 5.856(3) K at *f* = 0.5 Hz and *T*_SG_ = 6.075(1) K at *f* = 500 Hz, and these temperatures were obtained from fits by a Gauss function in the vicinity of *T*_SG_; the reported errors are merely errors of mathematical fits. The SG temperature shift of 0.012 is typical for SG materials [27,28].

Total specific heat data (*C*_p_) for the three compounds are shown in Figure 11 in the form of *C*_p_/*T* versus *T*. In the *x* = 0.25 sample, a λ-type anomaly near *T*_C_ proves the existence of long-range magnetic order. On the other hand, in the *x* = 0.50 sample, no clear λ-type anomaly near *T*_C_ is observed. This indicates that only a small amount of magnetic entropy is released at the magnetic ordering temperature because of significant dilution of the magnetic sublattices. In the *x* = 0.75 sample a specific heat anomaly appeared again near its *T*_SG_. As the three compounds are isostructural, they should have a similar lattice contribution to the total specific heat. Therefore, minimum total specific heat among the three samples in the temperature ranges of 20–40 K and 60–300 K could be taken as the lattice contribution with a smooth estimation between 40 K and 60 K. Below 20 K, the lattice contribution was estimated by a *βT*^3^ function passing through zero and *C*_p_ (*T* = 20 K, *x* = 0.25) points. It is interesting that in the A-site columnar-ordered Sm_2_MnMn(Mn_4−*z*_Ti*_z_*)O_12_, the opposite behavior of specific heat was observed, where the *z* = 1 sample showed no clear λ-type anomaly near *T*_C_, while the *z* = 2 sample demonstrated a clear λ-type anomaly near *T*_C_ [18].

The temperature dependence of the dielectric constant for the *x* = 0.75 and *x* = 0.50 samples is shown in Figure 12. The dielectric constant was found to be almost independent of temperature and frequency between 5 and 150 K for *x* = 0.50 with a value of about 32, and between 5 and 250 K for *x* = 0.75 with a value of approximately 50. For *x* = 0.50, the dielectric constant significantly increased above 150 K (especially at low frequencies). The same behavior was observed in many other related systems [16,18]. It is believed that the Maxwell-Wagner (extrinsic) polarization from increased conductivity is responsible for this behavior. No dielectric anomalies were observed near the magnetic phase transition temperatures. Increased conductivity in the *x* = 0.50 sample could be caused by electron hopping due to the presence of Mn in different oxidation states (+2 and +3). On the other hand, the *x* = 0.75 sample has only Mn^2+^ cations. Therefore, its resistivity increases, and that results in a frequency-stable dielectric constant in a wider temperature range.

In the parent solid solutions (Lu_1−*y*_Mn*_y_*)MnO_3_ (0.2 ≤ *y* ≤ 0.4), there is ferromagnetic ordering of Mn spins at the A and B sites and an AFM order between the two sites [10]. We suggest that a similar magnetic structure could be realized in (Lu_0.5_Mn_0.5_)(Mn_1−*x*_Ti*_x_*)O_3_ with *x* = 0.25 and 0.50 (Figure 3b), although this suggestion has yet to be confirmed by neutron diffraction. However, in this case, the B sublattice is diluted, and the ordered moment at the B site should therefore be noticeably reduced. When the ordered moment at the B site is small and saturated, that at the A site continues to increase with decreasing temperature. This scenario can of course explain why the magnetic susceptibility passes through a maximum and decreases again towards a lower temperature (Figure 5 and Figure 6c (where the maximum magnetization on the *M* versus *H* curves was observed at 60 K up to 70 kOe in the *x* = 0.25 sample)). In the case of (Tm_1−*y*_Mn*_y_*)MnO_3_, the total ordered moment at the A site exceeds the moment at the B site at a certain temperature (because of a large contribution from Tm^3+^), resulting in a negative magnetization phenomenon [11]. The absence of magnetic rare-earth cations in (Lu_0.5_Mn_0.5_)(Mn_1−*x*_Ti*_x_*)O_3_ prevented a negative magnetization.

All three samples have a Curie-Weiss temperature of the order of −100 K. This fact indicates that the strength of the average AFM interactions is almost the same, even for the highly diluted case *x* = 0.75 and suggests that Mn^2+^ cations at the A sites play an important role, as their amount is the same in all three samples. The *χ*^−1^ versus *T* curves of the *x* = 0.75 sample also demonstrated a noticeable deviation from the linear Curie-Weiss behavior below approximately 100 K, i.e., far above its *T*_SG_ = 6 K (Figure 8). Such deviations indicate the presence of strong ferromagnetic-like short-range spatial correlations [29].

The ideal Mn charge distributions should be the same in (Lu^3+^_0.5_Mn^2+^_0.5_)(Mn^3+^_0.50_Ti^4+^_0.50_)O_3_ and (Ca^2+^_0.5_Mn^2+^_0.5_)(Mn^3+^_0.5_Ta^5+^_0.5_)O_3_ [5]. However, there were differences in their magnetic properties. (Ca_0.5_Mn_0.5_)(Mn_0.5_Ta_0.5_)O_3_ showed a higher transition temperature of approximately 50 K, the absence of any magnetization decrease at lower temperatures, Curie-Weiss behavior in a wider temperature range down to approximately 80 K, a greater Weiss temperature of −260 K, and effective magnetic moment of 6.75 µ_B_/f.u. [5]. The larger transition and (absolute) Weiss temperatures of (Ca_0.5_Mn_0.5_)(Mn_0.5_Ta_0.5_)O_3_ could be explained by a larger ionic radius of Ca^2+^ in comparison with Lu^3+^ [2] resulting in a smaller (Mn/Ti)O_6_ octahedral tilt. It is a general tendency in perovskites that smaller octahedral tilts result in stronger exchange interactions (related to the Weiss temperature in the mean-field approximation) and higher magnetic transition temperatures [30]. Other differences in magnetic properties between (Lu_0.5_Mn_0.5_)(Mn_0.50_Ti_0.50_)O_3_ and (Ca_0.5_Mn_0.5_)(Mn_0.5_Ta_0.5_)O_3_ could be caused either by different magnetic structures or by the different temperature dependence of sublattice magnetizations; both of these can be determined by neutron diffraction in future works.

Ferrimagnetic transitions were found in this work in (Lu_0.5_Mn_0.5_)(Mn_1−*x*_Ti*_x_*)O_3_ with *x* = 0.25 and 0.50, where the R^3+^ and Mn^2+^ cations are disordered in one A site. Ferrimagnetic transitions also take place in A-site columnar-ordered quadruple perovskites R_2_MnMn(Mn_4−*z*_Ti*_z_*)O_12_ with *z* = 1 and 2, where R^3+^ and Mn^2+^ cations are ordered [18,19]. Therefore, when concentrations of magnetic cations are far above the percolation limits, magnetic properties do not depend qualitatively on ordered or disordered structural arrangements (but details of ferrimagnetic structures could, of course, be different). On the other hand, (Lu_0.5_Mn_0.5_)(Mn_1−*x*_Ti*_x_*)O_3_ with *x* = 0.75 shows spin-glass magnetic properties at a low temperature of 6 K while R_2_MnMn(Mn_4−*z*_Ti*_z_*)O_12_ with *z* = 3 exhibits long-range ferrimagnetic transitions at approximately 20–40 K [18,19,20]. Therefore, when concentrations of magnetic cations (at the B sites) are below the percolation limits, the structural order also supports long-range magnetic order.

## 4. Conclusions

Perovskite solid solutions, half-doped with magnetic Mn^2+^ cations at the A sites, were synthesized using a high-pressure, high-temperature method with the composition of (Lu_0.5_Mn_0.5_)(Mn_1−*x*_Ti*_x_*)O_3_ and *x* = 0.25, 0.50, and 0.75. Despite a significant dilution of A- and B-sublattices, the *x* = 0.25 and 0.50 samples show long-range ferrimagnetic order at *T*_C_ = 89 K and 36 K, respectively. The reduction in magnetization at low temperature suggests that both A and B sublattices are involved in the ferrimagnetic structures. The *x* = 0.75 sample exhibits a spin-glass transition at *T*_SG_ = 6 K, but it has a large Curie-Weiss temperature of −132 K and a large resultant frustration index of approximately 22.

## Figures and Tables

**Figure 1 materials-16-01506-f001:**
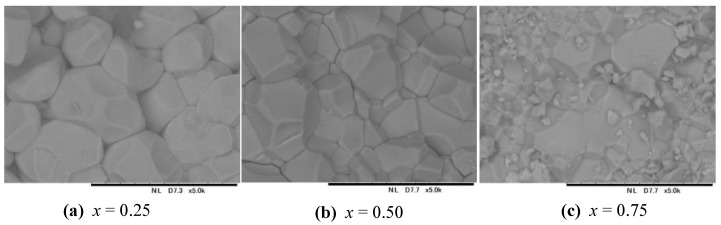
Scanning electron microscopy (SEM) images of the fractured surfaces of pellets of (Lu_0.5_Mn_0.5_)(Mn_1−*x*_Ti*_x_*)O_3_ with (**a**) *x* = 0.25, (**b**) 0.50, and (**c**) 0.75. The scale bar is 20 µm and magnification is 5000 on all panels.

**Figure 2 materials-16-01506-f002:**
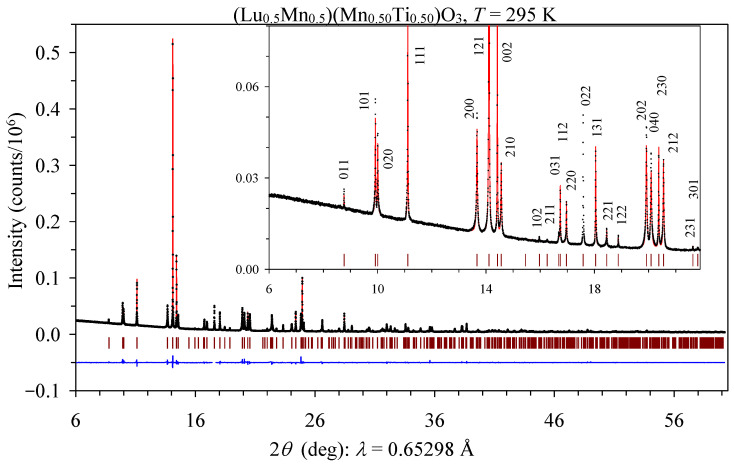
Rietveld refinement fits of room-temperature synchrotron X-ray powder diffraction data of (Lu_0.5_Mn_0.5_)(Mn_0.50_Ti_0.50_)O_3_. Black crosses: experimental data; red line: the calculated curve; blue line: the difference curve between the experimental and calculated points; brown tick marks: positions of possible Bragg reflections in the space group *Pnma*. The 2*θ* range is from 6° to 60.2°. One reflection near 17.6° was omitted from the refinement because of a poor agreement of the observed and calculated peak intensities, probably caused by a partial coarse-particle problem. Inset: a fragment of the observed and calculated data in a low 2*θ* range with Miller (*hkl*) indices of all observed reflections listed.

**Figure 3 materials-16-01506-f003:**
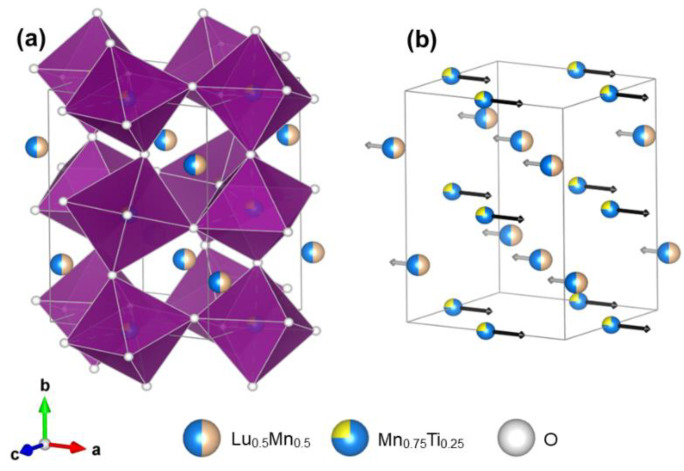
(**a**) Crystal structure of (Lu_0.5_Mn_0.5_)(Mn_0.75_Ti_0.25_)O_3_. (**b**) Possible ferrimagnetic structure of (Lu_0.5_Mn_0.5_)(Mn_0.75_Ti_0.25_)O_3_.

**Figure 4 materials-16-01506-f004:**
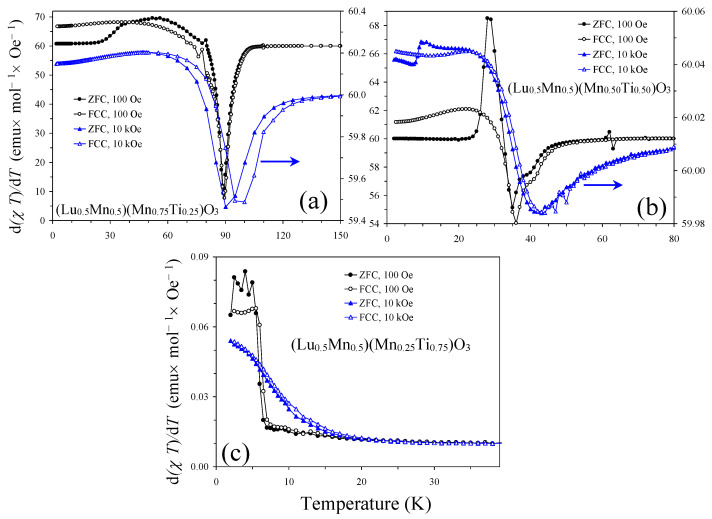
The ZFC and FCC d(*χT*)/d*T* versus *T* curves [ZFC: filled symbols, FCC: empty symbols] of (**a**) (Lu_0.5_Mn_0.5_)(Mn_0.75_Ti_0.25_)O_3_, (**b**) (Lu_0.5_Mn_0.5_)(Mn_0.50_Ti_0.50_)O_3_, and (**c**) (Lu_0.5_Mn_0.5_)(Mn_0.25_Ti_0.75_)O_3_ at 100 Oe (black circles) and 10 kOe (blue triangles). Magnetic transition temperatures were determined from peak positions on the 100 Oe FCC curves. All curves on (**a**,**b**) were shifted by +60 to produce positive values, as only relative values are important.

**Figure 5 materials-16-01506-f005:**
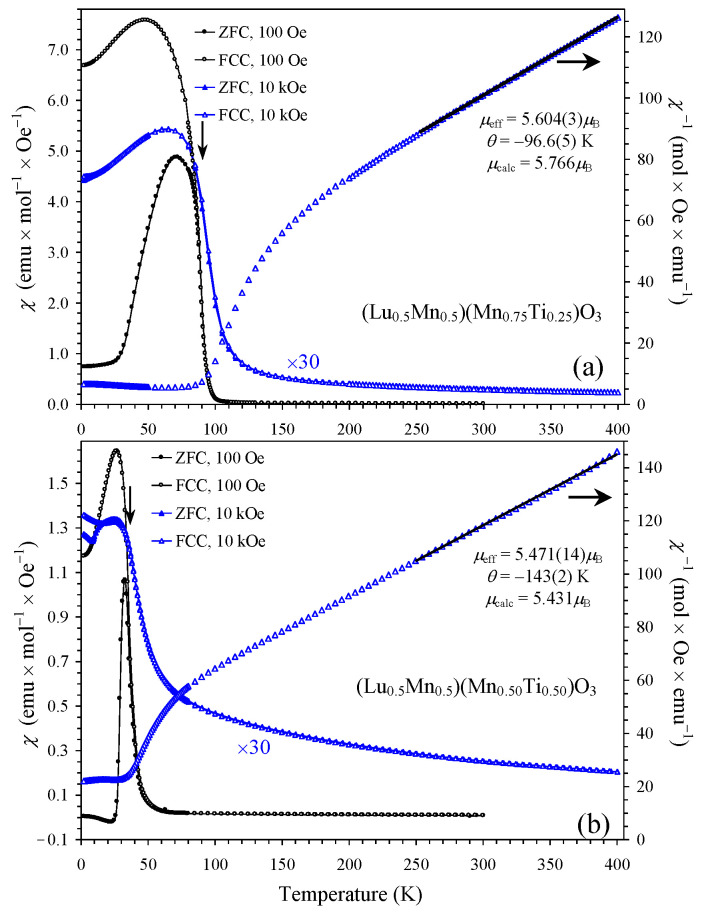
Left-hand axes: ZFC (filled symbols) and FCC (empty symbols) dc magnetic susceptibility curves (*χ* = *M*/*H*) of (**a**) (Lu_0.5_Mn_0.5_)(Mn_0.75_Ti_0.25_)O_3_ and (**b**) (Lu_0.5_Mn_0.5_)(Mn_0.50_Ti_0.50_)O_3_ at 100 Oe (black circles) and 10 kOe (blue triangles). The 10 kOe curves were multiplied by 30. Right-hand axes: *χ*^−1^ versus *T* curves (FCC, 10 kOe) with Curie-Weiss fits (black lines). Obtained fitting parameters are given in the figure. Arrows show magnetic transition temperatures.

**Figure 6 materials-16-01506-f006:**
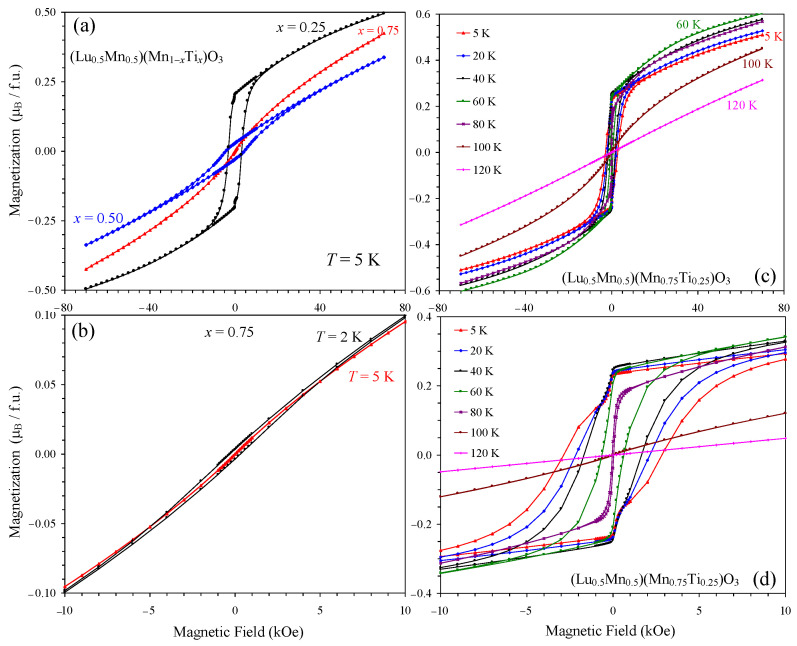
(**a**) *M* versus *H* curves of (Lu_0.5_Mn_0.5_)(Mn_0.75_Ti_0.25_)O_3_ (black circles), (Lu_0.5_Mn_0.5_)(Mn_0.50_Ti_0.50_)O_3_ (blue diamonds), and (Lu_0.5_Mn_0.5_)(Mn_0.25_Ti_0.75_)O_3_ (red triangles) at *T* = 5 K. f.u.: formula unit. (**b**) Enlarged fragment of the *M* versus *H* curves of (Lu_0.5_Mn_0.5_)(Mn_0.25_Ti_0.75_)O_3_ at *T* = 5 K (red triangles) and *T* = 2 K (black circles) at small magnetic fields. (**c**) *M* versus *H* curves of (Lu_0.5_Mn_0.5_)(Mn_0.75_Ti_0.25_)O_3_ at different temperatures. Panel (**d**) shows the details with enlarged scale for *x* = 0.25.

**Figure 7 materials-16-01506-f007:**
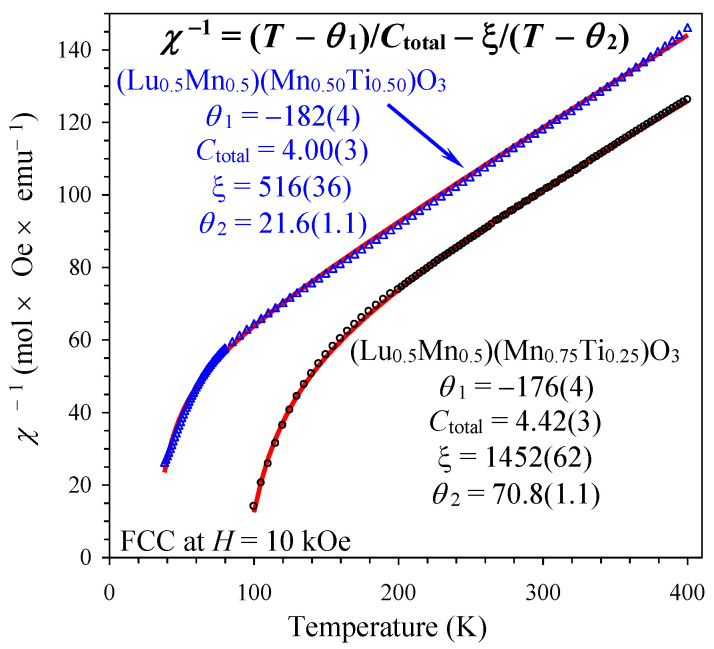
*χ*^−1^ versus *T* curves (FCC, 10 kOe) for (Lu_0.5_Mn_0.5_)(Mn_0.75_Ti_0.25_)O_3_ (black circles) and (Lu_0.5_Mn_0.5_)(Mn_0.50_Ti_0.50_)O_3_ (blue triangles) with fitting results (red lines) using a ferrimagnetic model [25,26]. The equation and obtained fitting parameters are given in the figure.

**Figure 8 materials-16-01506-f008:**
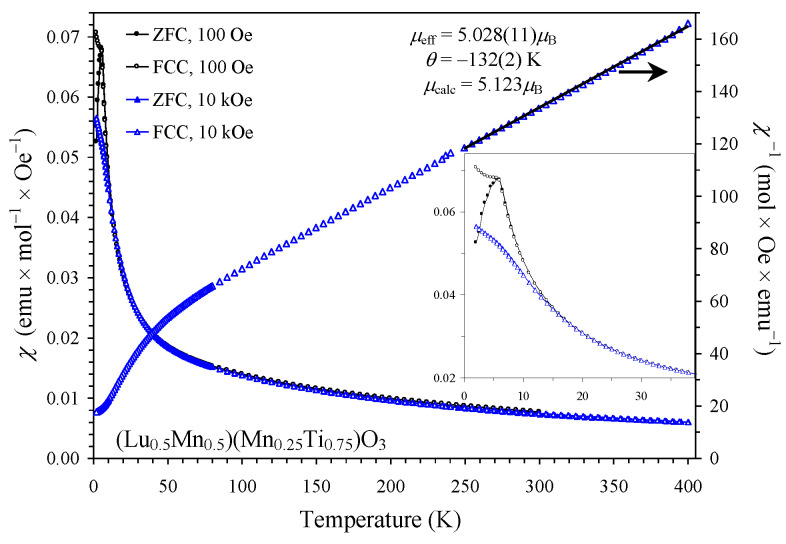
Left-hand axis: ZFC (filled symbols) and FCC (empty symbols) dc magnetic susceptibility curves (*χ* = *M*/*H*) of (Lu_0.5_Mn_0.5_)(Mn_0.25_Ti_0.75_)O_3_ at 100 Oe (black circles) and 10 kOe (blue triangles). The inset shows details below 40 K. Right-hand axis: *χ*^−1^ versus *T* curve (FCC, 10 kOe) with the Curie-Weiss fit (black line). Obtained fitting parameters are given in the figure.

**Figure 9 materials-16-01506-f009:**
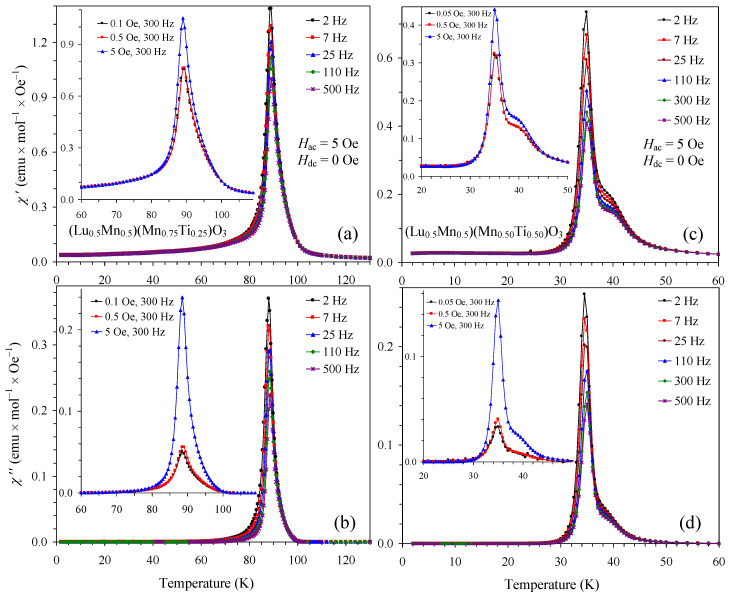
(**a**) Real χ′ versus *T* and (**b**) imaginary χ″ versus *T* curves of (Lu_0.5_Mn_0.5_)(Mn_0.75_Ti_0.25_)O_3_ at different frequencies. Insets show the χ′ versus *T* and χ″ versus *T* curves at different *H*_ac_ (0.1, 0.5 and 5 Oe) and one frequency (300 Hz). (**c**) Real χ′ versus *T* and (**d**) imaginary χ″ versus *T* curves of (Lu_0.5_Mn_0.5_)(Mn_0.50_Ti_0.50_)O_3_ at different frequencies. Insets show the χ′ versus *T* and χ″ versus *T* curves at different *H*_ac_ (0.05, 0.5 and 5 Oe) and one frequency (300 Hz).

**Figure 10 materials-16-01506-f010:**
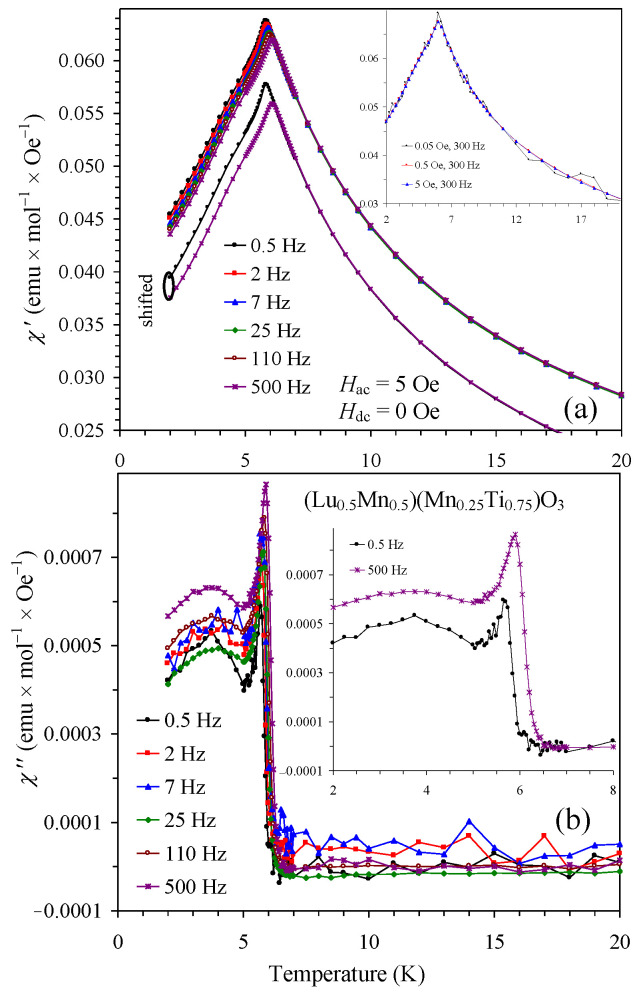
(**a**) Real χ′ versus *T* and (**b**) imaginary χ″ versus *T* curves of (Lu_0.5_Mn_0.5_)(Mn_0.25_Ti_0.75_)O_3_ at different frequencies. The (same) shifted curves at the smallest (0.5 Hz) and largest (500 Hz) frequencies are separately shown to clearly emphasize peak shifts. Insets: (**a**) χ′ versus *T* curves at different *H*_ac_ (0.05, 0.5 and 5 Oe) and one frequency (300 Hz). (**b**) χ″ versus *T* curves at the smallest (0.5 Hz) and largest (500 Hz) frequencies to clearly emphasize frequency-dependent shifts.

**Figure 11 materials-16-01506-f011:**
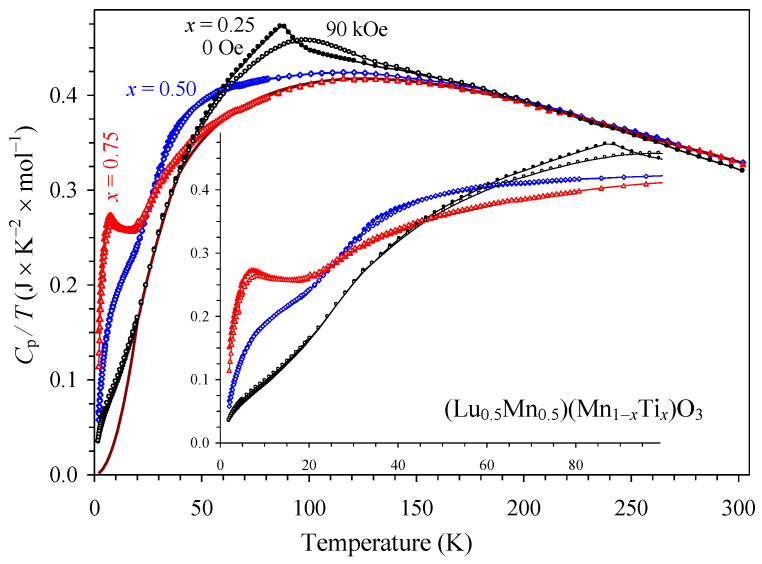
*C*_p_/*T* versus *T* curves of (Lu_0.5_Mn_0.5_)(Mn_0.75_Ti_0.25_)O_3_ (black circles), (Lu_0.5_Mn_0.5_)(Mn_0.50_Ti_0.50_)O_3_ (blue diamonds), and (Lu_0.5_Mn_0.5_)(Mn_0.25_Ti_0.75_)O_3_ (red triangles) at *H* = 0 Oe (filled symbols) and 90 kOe (empty symbols). The inset shows details below 100 K. *C*_p_ is the total specific heat. The brown line indicates the estimated lattice contribution to *C*_p_ (at all temperatures).

**Figure 12 materials-16-01506-f012:**
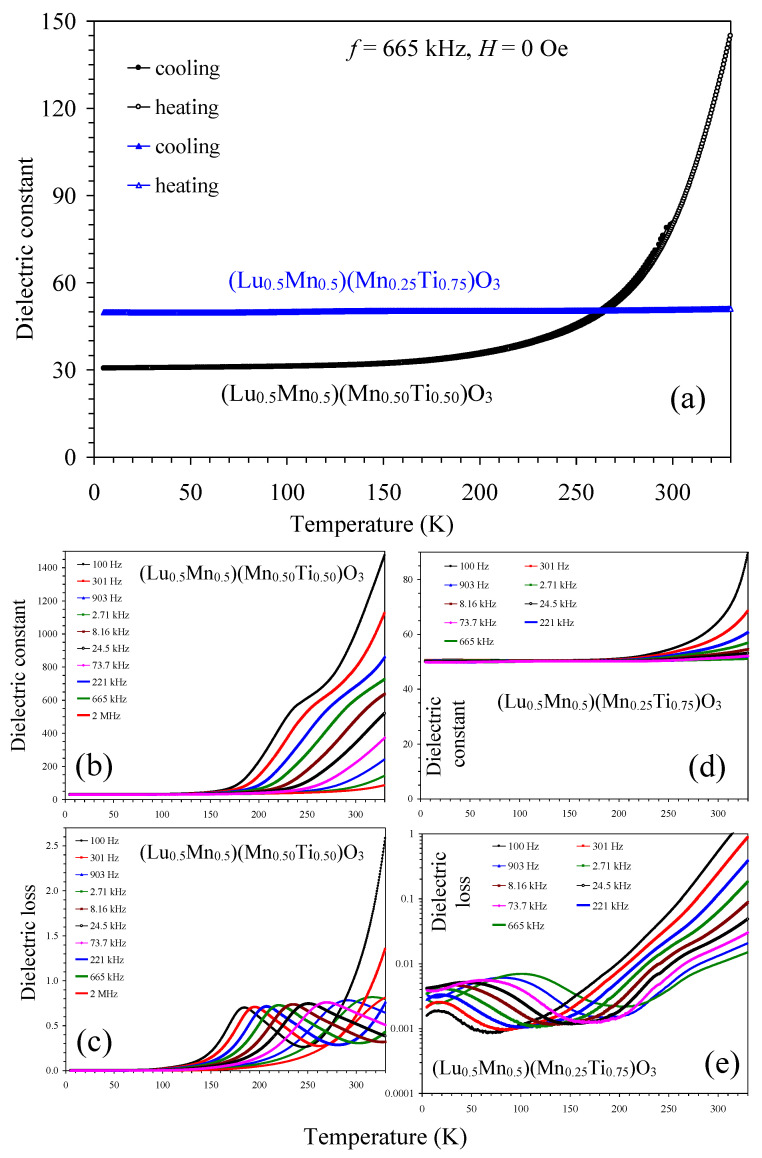
(**a**) The temperature dependence of the dielectric constant of (Lu_0.5_Mn_0.5_)(Mn_0.50_Ti_0.50_)O_3_ (black circles) and (Lu_0.5_Mn_0.5_)(Mn_0.25_Ti_0.75_)O_3_ (blue triangles) measured for the frequency *f* = 665 kHz at zero field (*H* = 0 Oe) for cooling and heating. (**b**,**c**) The temperature and frequency dependence of the dielectric constant and dielectric loss of (Lu_0.5_Mn_0.5_)(Mn_0.50_Ti_0.50_)O_3_ on heating. (**d**,**e**) The temperature and frequency dependence of the dielectric constant and dielectric loss of (Lu_0.5_Mn_0.5_)(Mn_0.25_Ti_0.75_)O_3_ on heating.

**Table 1 materials-16-01506-t001:** Structure parameters of (Lu_0.5_Mn_0.5_)(Mn_0.75_Ti_0.25_)O_3_, (Lu_0.5_Mn_0.5_)(Mn_0.50_Ti_0.50_)O_3_, and (Lu_0.5_Mn_0.5_)(Mn_0.25_Ti_0.75_)O_3_ at 295 K from synchrotron XRPD data.

Site	WP	*g*	*x*	*y*	*z*	*B*_iso_ (Å^2^)
(Lu_0.5_Mn_0.5_)(Mn_0.75_Ti_0.25_)O_3_						
Lu/Mn	4*c*	1	0.06601(7)	0.25	0.98503(11)	0.771(10)
Mn	4*b*	0.75	0	0	0.5	0.429(18)
Ti	4*b*	0.25	0	0	0.5	=*B*(Mn)
O1	4*c*	1	0.4570(7)	0.25	0.1064(8)	1.54(10)
O2	8*d*	1	0.3117(6)	0.0551(4)	0.6856(6)	1.41(7)
(Lu_0.5_Mn_0.5_)(Mn_0.50_Ti_0.50_)O_3_						
Lu/Mn	4*c*	1	0.06672(7)	0.25	0.98477(10)	0.694(10)
Mn	4*b*	0.5	0	0	0.5	0.528(18)
Ti	4*b*	0.5	0	0	0.5	=*B*(Mn)
O1	4*c*	1	0.4567(7)	0.25	0.1054(6)	0.74(8)
O2	8*d*	1	0.3145(5)	0.0583(4)	0.6910(5)	1.14(7)
(Lu_0.5_Mn_0.5_)(Mn_0.25_Ti_0.75_)O_3_						
Lu/Mn	4*c*	1	0.06407(5)	0.25	0.98450(8)	0.564(8)
Mn	4*b*	0.25	0	0	0.5	0.819(14)
Ti	4*b*	0.75	0	0	0.5	=*B*(Mn)
O1	4*c*	1	0.4458(5)	0.25	0.1193(4)	0.24(5)
O2	8*d*	1	0.3075(4)	0.0617(2)	0.6908(3)	0.63(5)

WP: Wyckoff position. *g* is the occupation factor. Space group *Pnma* (No 62); *Z* = 4. The Lu/Mn site has the occupation 0.5Lu+0.5Mn. (Lu_0.5_Mn_0.5_)(Mn_0.75_Ti_0.25_)O_3_: *a* = 5.44307(4) Å, *b* = 7.43580(3) Å, *c* = 5.18719(2) Å, and *V* = 209.944(2) Å^3^; *R*_wp_ = 5.09%, *R*_p_ = 3.11%, *R*_B_ = 2.20%, and *R*_F_ = 1.20%; *ρ*_cal_ = 6.838 g/cm^3^. No impurities. (Lu_0.5_Mn_0.5_)(Mn_0.50_Ti_0.50_)O_3_: *a* = 5.48737(2) Å, *b* = 7.48611(2) Å, *c* = 5.20269(1) Å, and *V* = 213.7211(11) Å^3^; *R*_wp_ = 3.62%, *R*_p_ = 2.33%, *R*_B_ = 3.06%, and *R*_F_ = 1.43%; *ρ*_cal_ = 6.662 g/cm^3^. No impurities. (Lu_0.5_Mn_0.5_)(Mn_0.25_Ti_0.75_)O_3_: *a* = 5.47716(2) Å, *b* = 7.60268(3) Å, *c* = 5.23395(2) Å, and *V* = 217.947(2) Å^3^; *R*_wp_ = 2.29%, *R*_p_ = 1.62%, *R*_B_ = 1.53%, and *R*_F_ = 0.88%; *ρ*_cal_ = 6.169 g/cm^3^. Lu_2_Ti_2_O_7_ impurity: 4.3 wt. %.

**Table 2 materials-16-01506-t002:** Selected bond lengths (in Å), bond angles (in deg), bond valence sums (BVS), and distortion parameters of Mn/TiO_6_ octahedra (Δ) in (Lu_0.5_Mn_0.5_)(Mn_0.75_Ti_0.25_)O_3_, (Lu_0.5_Mn_0.5_)(Mn_0.50_Ti_0.50_)O_3_, and (Lu_0.5_Mn_0.5_)(Mn_0.25_Ti_0.75_)O_3_.^a^.

	*x* = 0.25	*x* = 0.50	*x* = 0.75
Lu/Mn–O1	2.201(4)	2.216(3)	2.173(2)
Lu/Mn–O2 (×2)	2.191(3)	2.194(3)	2.206(2)
Lu/Mn–O1	2.219(4)	2.230(3)	2.206(3)
Lu/Mn–O2 (×2)	2.510(3)	2.499(3)	2.488(2)
Lu/Mn–O2 (×2)	2.583(3)	2.627(3)	2.697(2)
BVS(Lu^3+^)	3.00	2.93	2.94
BVS(Mn^2+^)	1.84	1.80	1.80
Mn/Ti–O1 (×2)	1.953(1)	1.965(1)	2.022(1)
Mn/Ti–O2 (×2)	1.969(3)	1.952(3)	1.988(2)
Mn/Ti–O2 (×2)	1.993(3)	2.038(3)	2.013(2)
BVS(Mn*^n^*^+^/Ti^4+^)	3.50	3.53	3.51
Δ(Mn/Ti)	0.7 × 10^–4^	3.7 × 10^–4^	0.5 × 10^–4^
Mn/Ti–O1–Mn/Ti	144.24(9)	144.58(9)	140.03(9)
Mn/Ti–O2–Mn/Ti (×2)	143.16(9)	142.65(9)	142.44(9)

^a^ BVS = $\sum_{i=1}^{i=N}\nu i$, *ν_i_* = exp[(*R*_0_ − *l_i_*)/*B*], *N* is the coordination number, *l_i_* is a bond length, *B* = 0.37, *R*_0_(Lu^3+^) = 1.971, *R*_0_(Ti^4+^) = 1.815, *R*_0_(Mn^2+^) = 1.79, *R*_0_(Mn^3+^) = 1.76, and *R*_0_(Mn^4+^) = 1.753 [24].

## Data Availability

Data are available from A.A.B. upon reasonable request.

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
