# Peer review of "Ferrimagnetic Ordering and Spin-Glass State in Diluted GdFeO3-Type Perovskites (Lu0.5Mn0.5)(Mn1−xTix)O3 with x = 0.25, 0.50, and 0.75"

_materials, 2023, doi:10.3390/ma16041506_

Round 1

Reviewer 1 Report

This paper deals with ferrimagnetic Ordering and Spin-Glass State in Diluted GdFeO3-type Perovskites (Lu0.5Mn0.5)(Mn1−xTix)O3 with x = 0.25, 0.50, 0.75.

The research described is performed at very high level. The style of presentation is clear and very informative.

The strengths of the presented work include the use of modern synthesis methods - high pressures and high temperatures, as well as a high level of characterization using synchrotron radiation. This, in particular, made it possible to collect manganese ions in three different oxidation states in one system. Of certain practical interest is the establishment of the fact that the permittivity of one of the samples is independent of temperature.

Author Response

Reviewer 1.

We thank the reviewer for positive evaluation of our work and recommendation to publish it as it is.

Reviewer 2 Report

1- Your have rich good data, but very poor discussions. 

2- The introduction must be rewritten to be more clear about the aim of your paper and the differences from the literature.

3- Nearly all figures need more discussions. Your paper seems to be only discriptive!

Author Response

Reviewer 2.

2- The introduction must be rewritten to be more clear about the aim of your paper and the differences from the literature.

Our reply.

We thank the reviewer for this suggestion. We extended the introduction part and added more ideas about the aim of this work (and 8 new references). The following paragraph was added into the revised manuscript.

“We found that the compositional range of (R1-yMny)MnO3 solid solutions can be extended to beyond y = 0.4 through an additional doping at the B sites in “double” solid solutions (R1-yMny)(Mn1-xTix)O3 [16]. For example, the doping level can reach y » 0.8 for y = x. There are several variable parameters for (R1-yMny)(Mn1-xTix)O3, such as, x, y, R and synthesis conditions. In this work, we report synthesis, crystal structures and physical properties of such “double” solid solutions with R = Lu and y = 0.5. We selected the half-doped level at the A sites (y = 0.5) in order to compare such perovskites, where R and Mn atoms are disordered in one site, with A-site columnar-ordered quadruple perovskites R2MnMn(Mn4-xTix)O12 (the generic composition is A2A¢A¢¢B4O12 [17]), where R and Mn atoms are ordered [18–20], to see effects of cation ordering on magnetic properties. In addition, we selected a non-magnetic R3+ cation to eliminate effects of rare-earth magnetism [16]. We observed that magnetic properties were qualitatively similar for x = 0.25 and 0.50. On the other hand, magnetic properties were quite different for x = 0.75, where structural disorder at the A sites produce spin-glass magnetic states while ordering of R and Mn atoms gives ferrimagnetic structures.”

1- Your have rich good data, but very poor discussions. and

3- Nearly all figures need more discussions. Your paper seems to be only discriptive!

Our reply.

We thank the reviewer for this suggestion. We extended the discussion parts and added the following.

“The ideal Mn charge distributions should be the same in (Lu3+0.5Mn2+0.5)(Mn3+0.50Ti4+0.50)O3 and (Ca2+0.5Mn2+0.5)(Mn3+0.5Ta5+0.5)O3 [5]. However, there were differences in their magnetic properties. (Ca0.5Mn0.5)(Mn0.5Ta0.5)O3 showed a higher transition temperature of about 50 K, the absence of any magnetization decrease at lower temperatures, the Curie-Weiss behavior in a wider temperature range down to about 80 K, larger Weiss temperature of -260 K and effective magnetic moment of 6.75µB/f.u. [5]. The larger transition and (absolute) Weiss temperatures of (Ca0.5Mn0.5)(Mn0.5Ta0.5)O3 could be explained by a larger ionic radius of Ca2+ in comparison with Lu3+ [2] resulting in a smaller (Mn/Ti)O6 octahedral tilt. It is a general tendency in perovskites that smaller octahedral tilts result in stronger exchange interactions (related to the Weiss temperature in the mean-field approximation) and higher magnetic transition temperatures [3]. Other differences in magnetic properties between (Lu0.5Mn0.5)(Mn0.50Ti0.50)O3 and (Ca0.5Mn0.5)(Mn0.5Ta0.5)O3 could be caused by either different magnetic structures or different temperature dependence of sublattice magnetizations; both of which can be determined by neutron diffraction in future works.

Ferrimagnetic transitions were found in this work in (Lu0.5Mn0.5)(Mn1-xTix)O3 with x = 0.25 and 0.50, where the R3+ and Mn2+ cations are disordered in one A site. Ferrimagnetic transitions also take place in A-site columnar-ordered quadruple perovskites R2MnMn(Mn4-xTix)O12 with x = 0.25 and 0.50, where R3+ and Mn2+ cations are ordered [18, 19]. Therefore, when concentrations of magnetic cations are far above the percolation limits magnetic properties qualitatively do not depend on ordered or disordered structural arrangements (but details of ferrimagnetic structures could, of course, be different). On the other hand, (Lu0.5Mn0.5)(Mn1-xTix)O3 with x = 0.75 shows spin-glass magnetic properties at low temperature of 6 K while R2MnMn(Mn4-xTix)O12 with x = 0.75 exhibits long-range ferrimagnetic transitions at about 20-40 K [18-20]. Therefore, when concentrations of magnetic cations (at the B sites) are below the percolation limit structural order supports long-range magnetic order as well.”

Reviewer 3 Report

Perovskite-type structures are currently of great interest due to the large number of potential applications. Thus, the object and subject of the manuscript presented are promising and up-to-date and fully correspond to the aims and scope of the journal. Belik et al. presented synthesis, crystal structures and physical properties of perovskite solid solutions (Lu1yMny)(Mn1xTix)O3. The research is well done and the paper is well organized. I recommend the submitted manuscript for publication after some revision.

1. Since the authors found that the range of compositions of solid solutions of (R1-yMny)MnO3 can be extended to the range of y = 0.4 and mentioned this, it will be useful to cite the corresponding reference.

2. In the "Introduction" section, the authors also reported the synthesis of the compounds studied. In this regard, I recommend providing SEM images of the samples as well as their chemical analysis for the benefit of the readers. In my opinion, the chemical analysis is necessary to confirm the occupancy of Lu and Mn in the A-site and to demonstrate that the Ti content in the B-site is as expected.

3. It would also be interesting to briefly discuss the reasons for the appearance of the Lu2Ti2O7 phase.

4. Please insert Tables 1 and 2 after their first mention.

5. Because of the resolution of the Figure 2 it is difficult to see the filled symbols and empty symbols. I recommend enlarging slightly the graphs shown in Figure 2.

Author Response

Reviewer 3.

  1. Since the authors found that the range of compositions of solid solutions of (R1-yMny)MnO3 can be extended to the range of y = 0.4 and mentioned this, it will be useful to cite the corresponding reference.

Our reply.

We thank the reviewer for this suggestion. We added a new reference [16].

  1. In the "Introduction" section, the authors also reported the synthesis of the compounds studied. In this regard, I recommend providing SEM images of the samples as well as their chemical analysis for the benefit of the readers. In my opinion, the chemical analysis is necessary to confirm the occupancy of Lu and Mn in the A-site and to demonstrate that the Ti content in the B-site is as expected.

Our reply.

We thank the reviewer for this suggestion. We performed new SEM/EDX experiments and added these results into the revised manuscript (a new Figure 1 shows the SEM images). The compositions were confirmed by the EDX measurements.

  1. It would also be interesting to briefly discuss the reasons for the appearance of the Lu2Ti2O7 phase.

Our reply.

We added a general discussion about reasons and effects of the appearance of impurities.

  1. Please insert Tables 1 and 2 after their first mention.

Our reply.

We moved Tables 1 and 2 to a different location as suggested.

  1. Because of the resolution of the Figure 2 it is difficult to see the filled symbols and empty symbols. I recommend enlarging slightly the graphs shown in Figure 2.

Our reply.

We increased the size of symbols on (old) Figure 2 (it is Figure 4 in the revised manuscript) and expanded the figure itself.

Reviewer 4 Report

In this paper, the authors have investigated the structure, magnetic and dielectric properties of perovskite (Lu0.5Mn0.5)(Mn1−xTix)O3 compounds with x = 0.25, 0.50, 0.75 by the XRD, Rietveld analysis of XRD, the measurements of magnetic properties, the specific heat and dielectric properties. The research technology is rich and advanced. However, some importance issues should be addressed before acceptance.

A very recently updated article [Yan Wang, Haiou Wang, Weishi Tan, Dexuan Huo, Journal of Applied Physics 132, 183907 (2022) ] related to the structure, spin glass state and magnetic properties of the perovskite oxides should be cited in the Introduction part.

The Miller indices of the diffraction peaks are suggested to be marked in Figure 1.

The authors claimed that “... there is ferromagnetic ordering of Mn spins at the A and B sites, and an AFM order between the two sites [7]. We suggest that a similar magnetic structure is realized in (Lu0.5Mn0.5)(Mn1−xTix)O3 with x= 0.25 and 0.50....”. The authors are suggested to draw the relevant magnetic structure diagram, that is more convenient for readers to understand.

In Figures 1, 2, 4 and 7, many data have overflowed the picture frame. The related picture is suggested to be redrawn.

Author Response

Reviewer 4.

  1. A very recently updated article [Yan Wang, Haiou Wang, Weishi Tan, Dexuan Huo, Journal of Applied Physics 132, 183907 (2022) ] related to the structure, spin glass state and magnetic properties of the perovskite oxides should be cited in the Introduction part.

Our reply.

The suggested reference was added into the revised manuscript.

  1. The Miller indices of the diffraction peaks are suggested to be marked in Figure 1.

Our reply.

(Old) Figure 1 was revised as suggested (it is Figure 2 in the revised manuscript).

  1. The authors claimed that “... there is ferromagnetic ordering of Mn spins at the A and B sites, and an AFM order between the two sites [7]. We suggest that a similar magnetic structure is realized in (Lu0.5Mn0.5)(Mn1−xTix)O3 with x= 0.25 and 0.50....”. The authors are suggested to draw the relevant magnetic structure diagram, that is more convenient for readers to understand.

Our reply.

We added a new Figure 3 with the crystal structure (panel (a)) and the proposed magnetic structure (panel (b)) for clearance.

  1. In Figures 1, 2, 4 and 7, many data have overflowed the picture frame. The related picture is suggested to be redrawn.

Our reply.

We modified some figures. However, some figures show zoomed parts, therefore, there should be overflowed data. Moreover, the look may depend on the program versions used.

Round 2

Reviewer 2 Report

- I still think that nearly all figures need more discussions. The paper seems to be only discriptive! The authors did not modulate that.

Author Response

We added some small additional discussion to some figures as the reviewer suggested. Unfortunately, the reviewer did not provide any details. Therefore, it is quite difficult to address such comments without clear instructions about what the reviewer wants to see.

We wanted to keep our paper clear and concise for the readers’ convenience. And other three reviewers already approved the manuscript as it was, supporting our way of the presentation of our results. We believe that we already discussed main points and conclusions that can be drawn from the figures. Of course, many additional details can be discussed. However, interested readers can see by themselves all the details from the figures shown.

Reviewer 3 Report

The authors have responded to all my comments. In my opinion, the manuscript can be published in the journal in its current form.

Author Response

We thank the reviewer for accepting our changes, supporting our way of the presentation of our results, and recommendation to publish as it is.

Reviewer 4 Report

The paper has been improved and can be accepted.

Author Response

(The authors gave the same response as above.)
